# "Faithful to What?" On the Limits of Fidelity-Based Explanations

**Jackson Eshbaugh**
Department of Computer Science
Lafayette College
Easton, PA 18042
eshbaugj@lafayette.edu

## Abstract

In explainable AI, surrogate models are commonly evaluated by their fidelity to a neural network's predictions. Fidelity, however, measures alignment to a learned model rather than alignment to the data-generating signal underlying the task. This work introduces the linearity score $\lambda(f)$, a diagnostic that quantifies the extent to which a regression network's input–output behavior is linearly decodable. $\lambda(f)$ is defined as an $R^2$ measure of surrogate fit to the network. Across synthetic and real-world regression datasets, we find that surrogates can achieve high fidelity to a neural network while failing to recover the predictive gains that distinguish the network from simpler models. In several cases, high-fidelity surrogates underperform even linear baselines trained directly on the data. These results demonstrate that explaining a model's behavior is not equivalent to explaining the task-relevant structure of the data, highlighting a limitation of fidelity-based explanations when used to reason about predictive performance.

## 1 Introduction

Despite the success of neural networks, understanding and trusting their learned input–output behavior remains a central challenge in interpretability. A common strategy in interpretability is to approximate a complex model with a simpler surrogate (Craven & Shavlik, 1995; Sato & Tsukimoto, 2001). This idea appears across paradigms: linear probes are introduced as tools to "better understand the dynamics" of a network and the role of intermediate layers (Alain & Bengio, 2018); LIME frames explanations as faithfully explaining individual predictions in an "interpretable manner," motivated by the need for trust (Ribeiro et al., 2016); and SHAP identifies a unique solution within additive feature-importance methods that satisfies desirable properties, suggesting "common principles about model interpretation" (Lundberg & Lee, 2017).

A common—but not guaranteed—implicit assumption arising from this framing is that if a surrogate is faithful to a model's behavior, then it should also capture much of the task-relevant structure responsible for predictive performance. Yet, fidelity measures agreement with a learned model's outputs rather than alignment with the data-generating signal; thus, a surrogate may explain what the network is doing while missing what matters for the task.

This paper formalizes this gap by introducing the linearity score $\lambda(f)$, defined as the coefficient of determination $R^2$ between a network's predictions and those of a linear surrogate trained to mimic them. Unlike interpretability techniques that produce local explanations or feature-level importance scores, $\lambda(f)$ provides a single, global measure of a network's *linear decodability*. We evaluate this framework on synthetic and real-world regression datasets, comparing baseline linear models, trained neural networks, and linear surrogates.

We find that surrogates can achieve high fidelity to a neural network while failing to recover the predictive gains that distinguish the network from simpler models, and in several cases substantially underperform linear baselines trained directly on the data. These results underscore the importance of distinguishing between explanations that faithfully model a network's behavior and explanations that capture the data's task-relevant structure.

## 2 METHODOLOGY

To assess whether a trained neural network's learned function is linearly structured, the degree to which its output behavior can be approximated by a linear model is measured. More precisely, the network is treated as a black-box function $f: \mathbb{R}^n \to \mathbb{R}$, and the degree of linear decodability of its input-output behavior is evaluated.

To begin, let $\mathcal{D} \subset \mathbb{R}^n$ be a domain over which the network operates, and assume $f \in L^2(\mathcal{D})$, the space of square-integrable functions. Let $\mathcal{L} \subset L^2(\mathcal{D})$ denote the subspace of affine functions. The optimal linear surrogate $g$ is then defined in Equation 1.

$$g = \arg \min_{g_i \in \mathcal{L}} \mathbb{E}_{x \sim \mathcal{D}} \left[ (f(x) - g_i(x))^2 \right] \tag{1}$$

Equation 1 defines the best linear approximation to the network's function $f$ in mean squared error. The **linearity score** $\lambda(f)$ is defined as the proportion of variance in the network's output that is captured by $g$:

$$\lambda(f) := R^2(f, g) = 1 - \frac{\mathbb{E}[(f(x) - g(x))^2]}{\text{Var}(f(x))} \tag{2}$$

$\lambda(f)$ exists in $(-\infty, 1]$, but values typically lie between 0 and 1. When $\lambda(f) \approx 1$, the network's output function is close to a linear subspace; when $\lambda(f) \approx 0$, it deviates substantially.

It is important to clarify that $\lambda(f)$ does not measure the linearity of the original data $(x, y)$, but rather the linearity of the network's *learned function* $f$ with respect to its input $x$. Even if the underlying data is highly nonlinear, a network may learn a function that is approximately linear—making its outputs linearly decodable. Conversely, a network may learn a highly nonlinear function even on linearly structured data. The optimal linear surrogate $g$ provides a closed-form approximation of $f$, but it is not guaranteed to preserve the components of the learned function that are responsible for predictive accuracy on the underlying task.

## 3 EXPERIMENTS

The proposed framework is evaluated in three stages: (i) a sanity check using a synthetic non-linear function, (ii) a real-world sanity check on the Medical Insurance Cost Dataset (Choi, 2017), and (iii) a realistic application case using the California Housing dataset (Kelley Pace & Barry, 1997) under distribution shift. These progressively demonstrate that: (i) $\lambda$ behaves sensibly as a measure of linear decodability, (ii) surrogate fidelity reflects alignment to a learned model rather than task-relevant structure in the data, and (iii) reasoning about predictive behavior from fidelity alone can be misleading in practice.

For each dataset, three models are evaluated[1]: (1) a baseline linear regression model trained directly on the input–output pairs; (2) a neural network tailored to the dataset (e.g., multi-layer perceptrons with varying layers, neurons, and ReLU activations, trained with Adam for 50–200 epochs—see Table 1; and (3) a linear surrogate model trained to approximate the output of the neural network. This setup allows the computation and comparison of the linearity score $\lambda(f)$ across diverse problem domains, and the assessment of how closely the neural network's behavior can be captured by a linear approximation.

---

[1]Experimental code is available on GitHub: `https://github.com/jacksoneshbaugh/lambda-linearity-score`

| Dataset | Network Architecture |
|---|---|
| Synthetic | $\mathbf{X} \to 64 \to \text{ReLU} \to 64 \to \text{ReLU} \to 1$ |
| Medical Insurance Cost | $\mathbf{X} \to 32 \to \text{ReLU} \to 16 \to \text{ReLU} \to 1$ |
| California Housing | $\mathbf{X} \to 128 \to \text{ReLU} \to 1$ |

Table 1: Architectures of the neural networks used in each experiment. Each sequence denotes the number of units, and activation functions.

| Dataset | $\lambda(f)$ | Baseline $R^2$ | Network $R^2$ | Surrogate $R^2$ |
|---|---|---|---|---|
| Synthetic (3.1) | -0.01 | $\approx$ -0.01 | 0.98 | $\approx$ -0.01 |
| Medical Insurance Cost (3.2) | 0.92 | 0.78 | 0.87 | 0.67 |

Table 2: Synthetic and Medical Insurance Cost dataset experimental results (reported in Sections 3.1 and 3.2).

While many surrogate methods, such as LIME (Ribeiro et al., 2016), are designed to provide local explanations, surrogate models are frequently used to reason about model behavior beyond strictly local neighborhoods, either through global surrogates, probes, or by aggregating local explanations. Our experiments focus on this broader use case by evaluating surrogate fidelity and predictive alignment across the data distribution. Results are presented in Table 2 and additional figures are available in Appendix A.

## 3.1 SYNTHETIC

As a sanity check, a synthetic regression task is constructed using

$$y = x \cdot \sin(x) + \epsilon, \text{ where } \epsilon \sim \mathcal{N}(0, 0.2^2) \text{ and } x \in [-4, 4]$$

A neural network fits the target function well (with $R^2 = 0.98$), while both a baseline linear model and a linear surrogate fail (both with $R^2 \approx -0.01$). Consistently, the linearity score is near zero ($\lambda(f) = -0.01$), indicating that the learned function is fundamentally non-linear and not linearly decodable. This confirms that $\lambda(f)$ behaves as expected.

## 3.2 MEDICAL INSURANCE COST

A real-world regression benchmark is evaluated using the Medical Insurance Cost dataset (Choi, 2017). A neural network outperforms a baseline linear model ($R^2 = 0.87$ vs. 0.78), indicating the presence of useful nonlinear structure in the data. However, a linear surrogate trained to mimic the network underperforms on the true target ($R^2 = 0.67$), despite achieving high fidelity to the network ($\lambda(f) = 0.92$).

This result highlights a key distinction: a linear surrogate that faithfully explains a neural network is not equivalent to a linear model that best explains the underlying data. Although the network's behavior is largely linearly decodable, the predictive gains that distinguish it from a linear baseline reside in components of the learned function that are not captured by the surrogate. As a consequence, explaining the model's outputs can obscure the task-relevant structure responsible for improved predictive performance.

## 3.3 CALIFORNIA HOUSING

Machine learning models are frequently evaluated under distribution shift, where explanations that appear reliable in-distribution may become misleading. To examine whether surrogate fidelity remains informative under distribution shift, the experimental procedure is refined and conducted on the California Housing dataset (Kelley Pace & Barry, 1997). Training is restricted to the middle 80% of income distribution and evaluation is IID within that split. Distribution shift is emulated by running the same tests OOD on the 10% lowest income homes and the 10% highest income homes.

| Domain | $\lambda(f)$ | RMSE($f$) | RMSE($g$) | $\Delta$RMSE |
|---|---|---|---|---|
| IID (mid-80%) | $0.651 \pm 0.005$ | $0.519 \pm 0.009$ | $0.718 \pm 0.000$ | $+0.199$ |
| Tail-L (bottom 10%) | $0.289 \pm 0.033$ | $0.548 \pm 0.007$ | $0.676 \pm 0.004$ | $+0.128$ |
| Tail-R (top 10%) | $0.675 \pm 0.007$ | $0.927 \pm 0.069$ | $0.841 \pm 0.004$ | $-0.086$ |

Table 3: California Housing results under distribution shift. $f$: neural network; $g$: linear surrogate trained to mimic $f$. $\Delta$RMSE = RMSE(g) − RMSE(f).

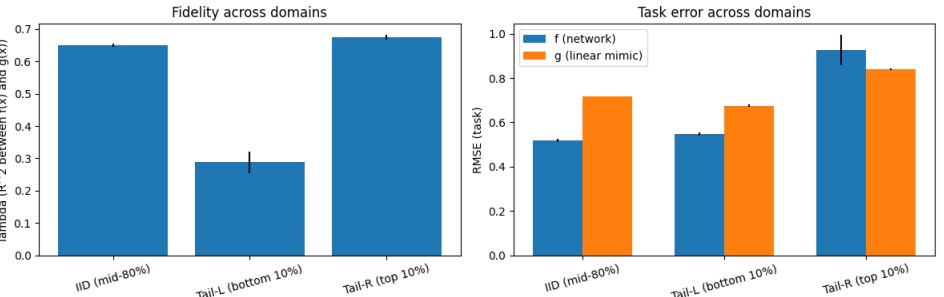

Figure 1: Results from the California dataset experiments.

Results are summarized in Table 3 and Figure 1. In the low-income tail, fidelity collapses ($\lambda(f) = 0.289$), reflecting that the network's behavior there is highly nonlinear and poorly captured by a linear surrogate. The IID and high-income regimes exhibit nearly identical fidelity ($\lambda(f) \approx 0.65$), yet the surrogate transitions from substantially worse than the network ($\Delta$RMSE = $+0.199$) to outperforming it ($\Delta$RMSE = $-0.086$).

This reversal highlights a fundamental distinction between explaining a model and explaining task-relevant behavior under distribution shift. Fidelity remains stable in the IID and high-income regimes because the surrogate continues to approximate the network's learned function, even as the network's alignment with the underlying task changes. As a result, high-fidelity explanations may fail to reflect—or may even invert—comparative predictive performance under shift. This finding underscores that surrogate fidelity measures alignment to a model, not robustness or reliability with respect to the data-generating process.

## 4 DISCUSSION & CONCLUSION

Across synthetic and real-world regression tasks, our results show that high surrogate fidelity does not necessarily track predictive performance. In multiple settings, linear surrogates closely approximate a network's output function while underperforming both the network and, in some cases, a baseline linear model trained directly on the data.

This pattern reflects a mismatch between what fidelity measures and how it is often interpreted in interpretability settings. Fidelity quantifies alignment to a learned model, whereas predictive accuracy depends on alignment with the data-generating signal. As a result, a surrogate may faithfully explain a model's behavior while failing to capture the task-relevant structure responsible for its predictive advantage—a failure mode that becomes especially pronounced when explanations are used to reason about generalization or robustness.

The linearity score $\lambda(f)$ therefore serves as a diagnostic of linear decodability, not as a guarantee of predictive usefulness or reliability. More broadly, these findings clarify the conditions under which explanation-based trust is warranted: fidelity-based explanations are informative about a model's behavior, but not necessarily about the structure that governs predictive performance. As surrogate-based explanations continue to be used in practice, treating fidelity as a proxy for task-relevant signal risks drawing misleading conclusions about model behavior.

ACKNOWLEDGEMENTS

I am grateful for the mentorship and guidance of Professors Jeffrey Pfaffmann, Jorge Silveyra, and Sofia Serrano. I am additionally grateful for the support of the Lafayette College Department of Computer Science.

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

## A    ADDITIONAL EXPERIMENTAL FIGURES

In this appendix, figures are provided for the synthetic and Medical Insurance Cost dataset experiments (Sections 3.1 and 3.2). Each figure provides plots of the true $y$ and predicted $y$ for the baseline regression model and neural network and a final plot of the neural network prediction against the surrogate (mimic) prediction. Figure 2 corresponds to the synthetic dataset and Figure 3 corresponds to the Medical Insurance Cost dataset.

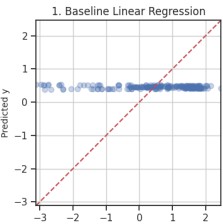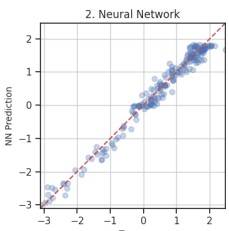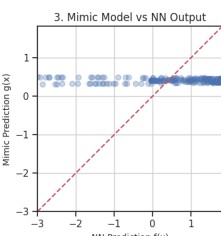

Figure 2: Predictions from the baseline linear model, neural network, and linear surrogate on the synthetic dataset.

Figure 3: Predictions from the baseline linear model, neural network, and linear surrogate on the Medical Insurance Cost dataset. Despite closely matching the network ($\lambda(f) = 0.9186$), the surrogate underperforms on the true target.

