# OpenReview forum: ""Faithful to What?" On the Limits of Fidelity-Based Explanations"
_ICLR.cc/2026/Workshop/Sci4DL — Sci4DL 2026_

### Official Review · Reviewer_SAvu · 2026-02-20

**Fit:** 2
**Significance:** 1
**Confidence:** 2

**Summary:**

This paper examines the issue of how surrogate models may be used in explaining ML models, and in explaining the characteristics of datasets.
Surrugate models may be trained to achieve a high-fidelity match for a network's output.

The authors introduce a metric called Lambda(f) which measures how well a linear regression model can match the network's predicitons when used as a surrogate model.

They show that a linear regression model can fit a network's predictions fairly accurately, while predicting the data less well than the network.
The authors identify that this discrepancy means that a good match between surrogate and model can be made even when the surrugate is making its predictions according to different rules.

**Strengths:**

1) Avoidance of confounding factors: Very simple datasets and architectures are used, so problems demonstrated here are likely to be problems in other settings
2) Code is made available to aid in reproducability

**Suggestions:**

1) Concise, Clear and Precise Research Question: It would help to be clearer about whether the paper is addressing problems with using surrogates to interpret models, or surrogates to interpret patterns in data. The final paragraph of the paper moves towards this, but being clear in the introduction or abstract would help.
2) Contextualisation of the field: It may help to extend the citations and discussion of related work to help clarify the area this paper pertains to, and motivate its relevance. For example, this paper seems potentially relevant: Méloux, M., Dirupo, G., Portet, F., & Peyrard, M. (2025). The Dead Salmons of AI Interpretability (arXiv:2512.18792). https://doi.org/10.48550/arXiv.2512.18792
Also "True to the Model or True to the Data" https://doi.org/10.48550/arXiv.2006.16234
3) Contextualisation of the current confusion: Can you cite references to current literature that conflate the issues.
3) Justification of Contributed Lambda score: The Lambda(f) linearity score is introduced as if it is a new research tool, but it was not obvious to me how it was generally useful beyond being an experiment for this paper. Could some suggestions be provided for how else it might be useful?

---

### Official Review · Reviewer_b3po · 2026-02-24

**Fit:** 3
**Significance:** 2
**Confidence:** 2

**Summary:**

The authors train linear models that best represent a dataset and use a new metric to compare this best performing model to the best linear approximation of a nonlinear model trained on that dataset. They call this metric a linearity score and show that the best linear representations of data are not necessarily the same as those for the best performing nonlinear model.

**Strengths:**

The core insight is well articulated/clear. I also liked the experimental progression from the synthetic sanity check, to a real-world dataset, and a distribution shift scenario. The linearity score is also easy to compute / interpretable by design.

**Suggestions:**

In the OOD case, separating standard OOD degradation vs fidelity limitation could be clearer/explored more.

Trying more architectures not just MLPs would be interesting

The introduction also invokes LIME, SHAP, and linear probes, suggesting broad implications for fidelity-based explanations as a class––these could be explored in a similar framework e.g. model vs data explanation.

---

### Official Review · Reviewer_MBoa · 2026-02-25

**Fit:** 2
**Significance:** 2
**Confidence:** 2

**Summary:**

This paper proposes a simple linearity score $\lambda(f)$ which quantifies the $R^2$ between the original model $f$ and a surrogate model $g$ trained on the outputs of $f$. The authors conduct experiments on 3 datasets, with 3 settings per dataset and find that although linear surrogates can have high fidelity (closely approximate the original model), this does not necessarily translate into high predictive performance. From 3.2 "a linear surrogate that faithfully explains a neural network is not equivalent to a linear model that best explains the underlying data". This is an interesting result that goes against a dominant paradigm in interpretability research where fidelity to the original model $f$ is one of the most important metrics.

**Strengths:**

- The paper is well written, clearly structured and easy to read.
- The linearity score is straightforward and easy to understand.
- The experimental setup makes sense. Starting with a toy dataset first and then progressively moving onto more difficult datasets. Training 3 models for each dataset also makes sense:  linear regression on the original data, NN on the original data, linear surrogate trained on outputs of NN.
- The results clearly demonstrate the overall claim that high fidelity of a surrogate model != high performance of surrogate model on original data for both the synthetic dataset and the medical insurance dataset. For the California housing dataset, splitting off the top/bottom tails of the population and evaluating them separately allows us to see the same issue for a more complex dataset.

**Suggestions:**

I don't have any major suggestions, the only question I have is about why $f$ is a neural network in this case? I believe all of these datasets are tabular, so wouldn't it make more sense to look at tree-based methods like XGBoost which work better on tabular data?

---

### Meta-Review · Area_Chair_W9fi · 2026-03-01

**Recommendation:** Accept

**Metareview:**

I recommend accept based on the reviews.

---

### Decision · Program_Chairs · 2026-03-02

Accept